# The Impact of Maternal Gut Microbiota during Pregnancy on Fetal Gut–Brain Axis Development and Life-Long Health Outcomes

**DOI:** 10.3390/microorganisms11092199

**Published:** 2023-08-31

**Authors:** Elizabeth M. Sajdel-Sulkowska

**Affiliations:** National Coalition of Independent Scholars (NCIS), Brattleboro, VT 05301, USA; e.sajdel-sulkowska@ncis.org or esajdel_sulk@alum.mit.edu

**Keywords:** fetal development, gut–brain axis (GBA), maternal gut bacteria, placenta

## Abstract

Gut microbiota plays a critical role in physiological regulation throughout life and is specifically modified to meet the demands of individual life stages and during pregnancy. Maternal gut microbiota is uniquely adapted to the pregnancy demands of the mother and the developing fetus. Both animal studies in pregnant germ-free rodents and human studies have supported a critical association between the composition of maternal microbiota during pregnancy and fetal development. Gut microbiota may also contribute to the development of the fetal gut–brain axis (GBA), which is increasingly recognized for its critical role in health and disease. Most studies consider birth as the time of GBA activation and focus on postnatal GBA development. This review focuses on GBA development during the prenatal period and the impact of maternal gut microbiota on fetal GBA development. It is hypothesized that adaptation of maternal gut microbiota to pregnancy is critical for the GBA prenatal development and maturation of GBA postnatally. Consequently, factors affecting maternal gut microbiota during pregnancy, such as maternal obesity, diet, stress and depression, infection, and medication, also affect fetal GBA development and are critical for GBA activity postnatally. Altered maternal gut microbiota during gestation has been shown to have long-term impact postnatally and multigenerational effects. Thus, understanding the impact of maternal gut microbiota during pregnancy on fetal GBA development is crucial for managing fetal, neonatal, and adult health, and should be included among public health priorities.

## 1. Introduction

Gut microbiota, a collection of bacteria, archaea, fungi, and viruses, is vital in maintaining our health. Gut microbiota is fully established after birth, although some limited microbial communities may be detected prenatally. Changes in gut microbiota occur over the life span and are specifically modified during life stages such as infancy, childhood, and puberty, when microbiota acquires sexually dimorphic characteristics [1,2], and aging [3,4].

The gut microbiota is also modified during pregnancy and nursing and is critical during fetal and infant development. Several recent studies have shown that the maternal gut microbiota during the early postnatal period affects the infant’s nervous system’s activity. However, more information is needed regarding the role of maternal gut microbiota during fetal development. Animal studies have demonstrated that maternal microbiota during pregnancy impacts fetal brain development and postnatal behavior [5,6,7]. Depletion of the maternal microbiota with antibiotics during pregnancy was shown to result in a deficiency of thalamocortical axons, impaired thalamic axon outgrowth in the fetus, and altered tactile sensitivity in adult offspring. In contrast, selective reconstitution of the maternal gut microbiome prevented abnormalities in fetal thalamocortical exogenesis [7]. A recent human study involving 1064 pregnant mothers and their 1074 children reported an association between the composition of maternal fecal microbiota during pregnancy and internalizing behavior associated with anxiety in children at the age of two years. Furthermore, fecal samples from pregnant mothers of children with normative behavior showed higher microbial alpha diversity and higher levels of butyrate-producing bacteria [8]. Thus, both animal and human studies point to an essential role of maternal microbiota during pregnancy in promoting fetal neurodevelopment.

Gut microbiota interacts with the brain via a bidirectional communication network, the gut–brain axis (GBA) that includes the central nervous system (CNS) and the autonomic nervous system (ANS) consisting of sympathetic, parasympathetic components including the vagus nerve (VN) and enteric nerves system (ENS), the immune system, the endocrine system including the hypothalamic–pituitary–adrenal (HPA), and the gut/gut microbiota. The bidirectional nature of the GBA network facilitates communication between the emotional and cognitive centers of the brain and the peripheral intestinal functions of the ENS [9].

Gut microbiota produces metabolites, hormones, and neurotransmitters, linking the gut with the brain. In turn, the brain impacts intestinal activities, including the activity of functional immune effector cells. Based on the dogma of a sterile womb [10], the predominant point of view stipulates that the offspring’s GBA is established postnatally and is regulated by the maternal gut microbiota delivered at birth. However, increasingly a small number of unique bacteria are being detected in fetuses that may represent transitional species facilitating the arrival of a full complement of microbiota after birth [11]. However, it is unlikely that prenatal gut microbiota can synthesize de novo metabolites and have an impact on the CNS development. It can be assumed that maternal microbiota metabolites and other factors are responsible for fetal development, including the GBA development. This assumption is supported by the observations that maternal gut microbiota dysbiosis during pregnancy is associated with abnormalities in the developing GBA gastrointestinal, neuronal, immune, and hormonal components, reviewed below.

This review presents the evidence supporting the prenatal impact of maternal gut microbiota on fetal GBA development. It focuses on the fetal development of the core components of GBA, the CNS, the VN, the immune system, the HPA, and the gut/ENS [12]. It further examines the supportive role of the placenta in regulating maternal–fetal communication and fetal development. Furthermore, it discusses the factors affecting gut microbiota during pregnancy and contributing to maternal gut dysbiosis, such as maternal obesity, diet, maternal stress and depression, infection, and drugs (antibiotics and antidepressants) that alter gestational GBA development. Finally, it alludes to the concept of “Baker’s hypothesis” or the developmental origins of health and disease (DOHaD) and the multigenerational impact of maternal gut microbiota dysbiosis during pregnancy.

## 2. Adaptation of Maternal Gut Microbiota to Pregnancy and Fetal Development

The maternal organism during pregnancy undergoes significant anatomical and physiological changes designed to support fetal nutritional and energy demands, regulate fetal growth and differentiation, and assure immunological tolerance. These changes affect every organ system in the body and metabolic, endocrine, immune, and neuronal functions [13]. In humans, the changes begin early in the first trimester, peaking at the time of labor and reverting to prepregnancy levels by a few weeks into the postpartum. Metabolic changes during pregnancy involve higher accumulation of fat and cholesterol due to increased leptin, insulin, and insulin resistance, resulting in pregnant women gaining more fat and cholesterol. During pregnancy, there is an increase in proinflammatory gut bacteria associated with increased levels of proinflammatory cytokines. These changes further improve energy storage in fat, providing for fetal growth.

Healthy human gut microbiota is dominated by phyla *Firmicutes*, *Bacteroidetes*, *Actinobacteria*, *Proteobacteria*, and *Fusobacteria*. The Firmicutes phylum comprises genera such as *Lactobacillus*, *Bacillus*, *Clostridium*, *Bacillus*, *Enterococcus*, and *Ruminicoccus*. *Bacteroidetes* consist of predominant genera such as *Bacteroides* and *Prevotella*. The less abundant Actinobacteria phylum is mainly represented by the *Bifidobacterium* genus [14].

Starting during the second trimester and increasing during the third trimester, the metabolic and immune changes observed in pregnant women are accompanied by changes in maternal gut microbiota [15]. These changes are characterized by a transition to a low alpha diversity (low richness of taxa and abundance taxa) and a high beta index (high variability in composition) associated with increased glycogen- and lactose-producing bacteria, a reduction in butyrate-producing bacteria [16] and lower diversity, but enrichment in Actinobacteria and Proteobacteria phyla. Levels of *Faecalibacterium*, a member of *Firmicutes* phylum, a butyrate-producing bacterium with anti-inflammatory activities, significantly decrease in the third trimester of pregnancy. Beta diversity increases in the third trimester, and is coupled with weight gain, insulin insensitivity, and higher levels of fecal cytokines, reflecting inflammation [14,17,18]. There is an increase in Firmicutes, associated with an increase in the need for energy storage, and in *Proteobacteria* and *Actinobacteria* levels that have a protective effect on both the mother and the fetus via proinflammatory mechanisms [14]. These changes in maternal gut microbiota meet a metabolic demand of the fetus [19], contribute to the increase in fetal body weight, and provide glucose, but they result in maternal hyperglycemia.

Furthermore, during normal pregnancy, the maternal gut functions and bacterial composition are affected by the inflammatory and immune changes necessary for supporting pregnancy and brought about by altered hormonal changes, notably pregnancy-specific hormone, human chronic gonadotrophin (hCG). In turn, hCG regulates the secretion of estrogen and progesterone, which impacts maternal gut microbiota composition. Elevated progesterone levels increase gastrointestinal transit time, a critical factor in shaping gut microbiota composition and activity [20].

The changes in maternal gut microbiota during healthy pregnancy are essential to support maternal health and fetal development. Complicated pregnancies are associated with altered maternal gut microbiota composition, possibly related to the rise in progesterone levels [21]. Furthermore, microbial signatures have been linked to embryonic development, imprinting the CNS and the immune system and fine-tuning the balance between health and disease.

Microbial gut dysbiosis is an imbalance in microbiota homeostasis and affects health status. During pregnancy, maternal gut dysbiosis refers to a disruption of an adaptation of the microbiota to pregnancy conditions associated with maternal status and presenting the risk to the mother and the fetus. Microbial gut dysbiosis in pregnant women can be caused by maternal obesity, diet, stress, inflammation, infection, antibiotic, and antidepressant use. Obesity during pregnancy is associated with an increase in *Firmicutes*, resulting in a high gut *Firmicutes*-to-*Bacteroidetes* ratio. The increase in *Firmicutes* may increase calorie absorption, predispose to weight gain, and is related to gut inflammation and intestinal permeability. Furthermore, obesity during pregnancy can lead to a change in bacterial phyla observed in nonobese pregnancies [22]. A detailed discussion of microbial changes associated with maternal gut dysbiosis is beyond the scope of this review.

## 3. Maternal Gut Microbiota–Fetal Interaction during Pregnancy: Role of the Placenta

The interaction between the maternal gut microbiota and the fetus remains a subject of intense debate centered around the dogma of a “sterile womb”, postulating no contact between maternal gut bacteria and the fetus and, instead, fetus dependence on the metabolites derived from the maternal gut microbiota. More recent evidence suggests a maternal gut microbiota translocation to the uteroplacental unit [23].

Irrespective of the microbiota present in the uterus, maternal gut microbiota metabolites supply energy, nutrients, and vitamins such as B complex, folate, choline, betaine, and short-chain fatty acids (SCFAs) derived from the dietary fiber fermentation, polyphenols that provide nutritional programming of fetal growth and development via epigenetic mechanisms, and neurotransmitters. Maternal metabolites pass from the gut lumen to the circulation, access the fetus through the placenta, reach fetal circulation, and provide the nutrients required for fetal growth and development. These nutrients also impact gene expression and brain development [24]. Some metabolites also cross the fetal blood–brain barrier (BBB) and are involved in fetal CNS and immune system development by regulating dendritic T-cell functions and cytokine production [25].

The placenta is a critical structure for normal fetal development. The placenta forms a primary barrier between the maternal environment and the fetus, regulating gas exchange, hormone production, and secretion of nutrients. In healthy pregnancies, the placenta also harbors microbiota. It has been speculated that the placental microbiota may contribute to fetal immune system development during the second trimester of pregnancy [26]. The placental microbiota is unique, resembles maternal oral microbiota and can adversely affect pregnancy outcomes [27]. Interestingly, placental membranes defend the fetal environment and have bactericidal properties as they contain cells, extravillous trophoblasts, natural killer cells, leukocytes, and macrophages. Although the bacteria can pass the barrier due to bacterial ligands, they are most likely not viable and fragmented. It is also possible that specific microorganisms may hide inside placental trophoblasts [28].

In mice, fetal placental vascularization occurs between gestation days 8–11 when the placenta gains contact with maternal circulation and transport from the maternal circulation to the fetus is initiated. Metabolites of maternal gut microbiota, such as SCFAs, are transferred to the liver, enter the bloodstream, pass across the placental barrier, enter the fetal circulation, and contribute to the formation of BBB and innate immune development. Maternal stress during the first week of pregnancy in mice induces rapid and lasting alterations in maternal gut microbiota resulting in alterations of placental transfer of nutrients in a sex-specific manner where male but not female offspring demonstrate significant neurodevelopmental changes in the hypothalamic and limbic circuit and the regulation of stress responsivity [29]. These findings indicate that the composition of maternal gut microbiota during pregnancy is a crucial contributor to the metabolic programming of offspring [30].

While the fetus is initially protected from harmful substances in maternal blood by the placenta, subsequent maturation of the BBB provides a second specific safeguard against pathogens and potentially toxic substances critical for healthy brain development during pregnancy. In rodents, BBB is formed at gestational day 13.5–15.5, but the early BBB differs from adults with a higher concentration of the tight junctions (TJ proteins) and extracellular matrix components. In germ-free mice, the BBB is leaky at gestational day 16.5, suggesting that the maternal microbiota is critical for developing a functional BBB [31]. In humans, some efflux transporters that can exclude toxins are already present at eight weeks of gestation [32], and BBB components are present at 12 weeks of gestation. It is widely believed that during gestation and in newborns, this barrier is immature or “leaky”, rendering the developing brain more vulnerable to drugs or toxins entering the fetal circulation from the mother [25].

It has been suggested that increased intestinal permeability in early pregnancy is associated with increased maternal levels of LPS, and cytokines at the endometrial level, facilitating the translocation of bacterial metabolites and bacteria from the intestinal lumen into maternal circulation [16]. Some bacteria may travel to the placenta via the bloodstream after gut epithelium translocation by dendritic cells as they migrate to lymphoid organs. In addition, maternal oral bacteria entering the bloodstream have also been suggested as a source of fetal microbiota. This early prenatal microbiota may imprint immune systems in preparation for postnatal microbiota seeding. Following birth, the human intestine is rapidly colonized by microbiota, mostly strict anaerobes. By the end of the first year, infants possess an individually distinct microbiota; by 2–5 years, the microbiota fully resembles adults in terms of composition and diversity [33].

The initial diffusion of maternal metabolites across the placenta facilitates the fetal nervous system and HPA axis development. It is followed by the infiltration of gut bacteria via the placenta and translocation into the fetal circulation. Thus, gut microbiota colonization may begin during fetal life [34,35]. Some bacteria may infiltrate the fetal intestine, but they are insufficient and very distinct to trigger the activation of the intestinal epithelium tolerance that occurs during the first contact with microbiota after birth.

## 4. Effect of Maternal Gut Microbiota during Pregnancy on Fetal GBA Development

While there is much research on the role of gut microbiota and GBA during the postnatal period [36], little is known about gut microbiota and GBA during the prenatal period. Because of the fetal developmental trajectory, there is no functional GBA during fetal life. In humans, the rudimentary GBA development during the fetal period is regulated by maternal gut microbiota.

At birth, massive amounts of maternal vaginal, fecal, and skin microbiota colonize the emerging rudimentary gut of the newborn delivered by the vagina; newborns delivered by C-section are exposed to the skin microbiota. Additional maternal microbiota is transferred during nursing; the composition of microbiota transferred via milk is affected by maternal health and gestational age. Gut closure around six months marks the formation of functional GBA [37]. Functional GBA consists of a bidirectional network involving the CNS, ANS, ENS, VN, neuroendocrine and neuroimmune systems, the HPA, and the gut/gut microbiota. The microbiota is a powerhouse of the GBA, providing bioconversion of nutrients, detoxification, regulation of immunity, and protection against pathogenic microbes.

Accumulating preclinical and clinical data suggest the critical role of maternal gut microbiota in regulating fetal development by transferring maternal metabolites and other factors to the fetus via the placenta. So, while fully functional bidirectional offspring GBA develops postnatally, the rudimentary GBA is assembled under the maternal-gut microbiota regulation during the fetal period.

### 4.1. Impact of Maternal Microbiota during Pregnancy on Fetal CNS

Maternal gut microbiota during pregnancy is crucial for fetal CNS development. Additionally, other factors such as maternal genetics, diet, health status, stress, and medication contribute to the outcome [38].

The link between maternal gut microbiota and fetal neurodevelopment is based on animal and human studies. Rodent studies have shown gender-dependent abnormal fetal brain development in germ-free mice [39]. The microbiota-deficient mice showed altered gene expression involved in neurotransmission, neuroplasticity, metabolism, and morphology in the hippocampus [23] and thalamocortical neurodevelopment linked to sensorimotor behavior and pain perception postnatally [7].

The human brain and nervous system begin to develop during the embryonic period starting at six weeks and continue through the end of pregnancy, with development proceeding postnatally into puberty and beyond. Structural and functional, time-specific neurodevelopmental changes occur during gestation, including axonal growth, synapse formation, and dendritic and axonal arborization, followed by synaptic connections. Maternal gut microbiota during pregnancy regulates metabolites that reach the fetus via transplacental signaling and enter the fetal brain [7]. Maternal gut microbiota promotes healthy fetal brain development and affects structural and functional brain connectivity, impacting the offspring’s cognitive and behavioral development. Clinical studies suggest maternal gut microbiota dysbiosis during pregnancy impacts the fetal CNS’s physiological and functional development. Specifically, microbial depletion due to infection or antibiotic treatment during pregnancy alters fetal neurodevelopment, contributing to abnormal brain structure and function underlying maladaptive, autism-like behaviors in offspring (Figure 1) [6,7,40,41]. Depletion of maternal gut microbiota during pregnancy also affects gene expression in the developing fetal brain, including genes regulating new axons expression connecting the thalamus with the cortex responsible for sensory functions.

In addition to metabolites, gut microbiota produces neurotransmitters and neuromodulators, serotonin (or 5-hydroxytryptamine (5-HT)), gamma-aminobutyric acid (GABA), and SCFAs. These bioactive factors are transported to the fetal brain via blood vessels after crossing the placenta and BBB. The placenta is enriched with a complex vascularization for fetal blood supply that requires extensive angiogenesis. During pregnancy, maternal gut microbiota produces SCFAs involved in neuronal, intestinal, and pancreatic differentiation, facilitated by G protein-coupled receptors. Maternal gut metabolites also promote fetal thalamocortical exogenesis. A region-specific change in the neurotransmitter system was noted in the brains of germ-free mice that showed increased serotonin (5-HT) concentration in the hippocampus [39].

Notably, the placenta synthesizes 5-HT that impacts fetal CNS development by regulating cell proliferation, migration, and wiring during prenatal development. It has been shown that the 5-HT precursor, tryptophan, is derived from maternal gut metabolites. Placental 5-HT reaches the fetal forebrain during cortical neurogenesis and the initial axon growth period. In mice, serotonergic neurons appear in the fetal mouse hindbrain on E10.5, and by E14.5, these neurons reach the neocortex; 5-HT signaling is also vital for the thalamocortical axons. Chronic mild stress during pregnancy in rats increases free tryptophan concentration in the maternal blood and the fetal brain, increasing anxiety in the offspring. In humans, placental synthesis of 5-HT occurs during the first and second trimesters of pregnancy [42]. Disruption of placental 5-HT signaling results in long-term behavioral abnormalities such as anxiety postnatally. Maternal gut microbiota also affects the development of fetal BBB. Animal studies have shown increased BBB permeability in GF mice, associated with lower brain innate immunity and reduced thalamocortical axon growth [43].

**Figure 1 microorganisms-11-02199-f001:**
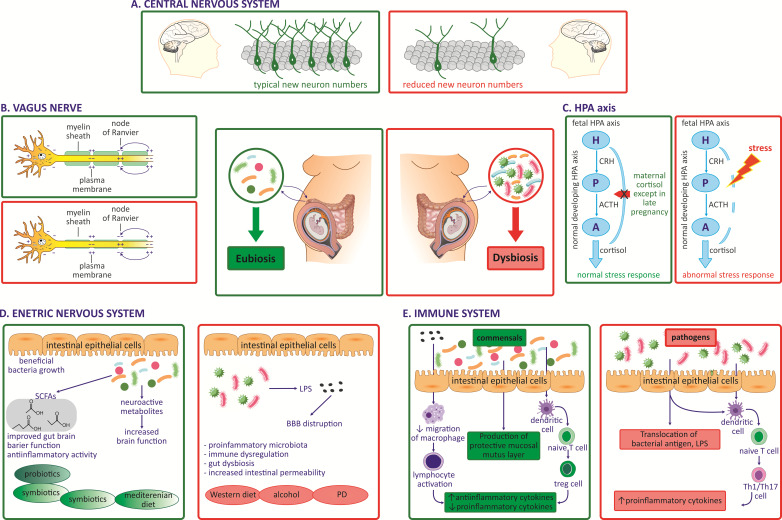
Impact of maternal gut microbiota dysbiosis during pregnancy on fetal gut–brain axis development. (**A**) Reduced brain neurogenesis [40,44]. (**B**) Reduced myelination of the vagus nerve impacts its maturation [36,40,45,46]. (**C**) Hyperactivation of HPA axis [47,48]. (**D**). Deficiency in utero SCFAs impacts fetal ENS development [49,50,51,52]. (**E**) Maternal gut dysbiosis impacts fetal immune system and fetal regulatory T (Tregs) cells [11,53]. (Selected examples of mostly clinical studies).

### 4.2. Impact of Maternal Microbiota during Pregnancy on the Fetal VN

The VN innervates the intestine from the proximal duodenum to the distal descending colon. It is a major bidirectional highway between the gut and the brain, specifically the nucleus of the solitary tract (NTS) and, in turn, to the paraventricular nucleus of the hypothalamus (PVN) and arcuate nucleus communicating gut’s status in terms of chemical content distension, and inflammation [54]. Maternal gut microbial metabolites, such as SCFAs, are transferred across gut epithelium via specific transporters and activate VN. Afferent VN fibers transport signals from the gut microbiota to the brain. In response to these stimuli, the brain sends signals back to entero-epithelial cells via efferent VN fibers [55].

VN sensory and motor nuclei have been observed in E11–E14 mice embryos [56]. In humans, morphological studies demonstrate rapid developmental increase in the total number of myelinated vagal fibers from 24 weeks through adolescence, with the most significant increases observed from approximately 30–32 weeks of gestation to approximately six months postpartum [57]. An increase in sympathetic activation, associated with higher parasympathetic modulation and baseline stabilization ability, is reached during the transition from the late second to the early third trimester [58].

Myelination of vagal efferent fibers related to cardiac functions begins during the last trimester [59]. Maternal gut dysbiosis induces microglial activation and leads to systemic neuroinflammation [45]. While eubiosis promotes myelination [36], dysbiosis-associated increase in proinflammatory cytokines can cause direct injury to the differentiating neurons and oligodendrocytes at a critical period of active myelination supporting the direct impact of maternal gut dysbiosis-induced inflammation on fetal brain development [40]. A fetal inflammatory response syndrome (FIRS) characterized by elevated fetal proinflammatory cytokines and hypoxic–ischemic events has been observed in preterm infants and is accompanied by myelination failure [60].

### 4.3. Impact of Maternal Microbiota during Pregnancy on Fetal Immune System Development

Both rodent and human studies support the involvement of maternal microbiota during pregnancy in the development of fetal innate and adaptive immunity [11]. Studies in germ-free pregnant mouse dam transiently colonized with *E. coli* showed altered innate lymphoid cells in pups. Additionally, mouse pups derived from dams treated with antibiotics during pregnancy had altered innate immunity at 14 days postpartum [61]. Furthermore, exposure of nonobese diabetic (NOD) mice to vancomycin during pregnancy increased offspring susceptibility to Type 1 diabetes (T1D) [53].

In mice, a limited number of pre-B cells were found in bone marrow by gestation day 19, and T lymphocytes were detected by birth [62]. In humans, the B- and T-cells are found between 12 and 14 weeks of gestation and increase until birth. Both innate and adaptive immunity is present by 16 weeks of gestation. The first set of lymphoid tissues, the mesenteric lymph nodes, and Peyer’s patches develop in the sterile environment of the fetus [63].

Overall, the immaturity of the immune system is paralleled by the immaturity of the intestinal barrier function, resulting in a higher passage of antigens across the intestine. Importantly, in humans, during fetal life, mothers transfer natural passive immunization. Microbial antigens derived from maternal gut microbiota form complexes with maternal antibodies (antigen-IgG) and are transferred via the placenta and induce immune activation in the fetus. Free dietary antigens may also pass the placental barrier [64]. During the fetal period, intestinal digestion is low, and components in the amniotic fluid, like proteinase inhibitors, influence the luminal environment and formation of antigenic molecules. It has been suggested that maternal microbiota metabolites are critical for immune functions [65]. SCFAs produced by maternal microbiota during pregnancy have been shown to regulate intestinal immunity, T-cell development, dendritic cells (DCs) activity, and epithelial integrity [11]. Furthermore, specific neuropeptides, vasoactive intestinal peptides, and norepinephrine modulate the functions of dendritic cells and cells located throughout the wall of the intestine and secondary lymphoid tissues like Peyer’s patches. Additionally, it has been suggested that bacterial DNA transferred from the mother to the fetal gut stimulates mucosal immune development [66] and impacts the fetus’s immune system in preparation for postnatal maternal–fetal microbiota transfer.

In germ-free mice, fetal thymic CD4+ T cell and Treg cell development are compromised but rescued by maternal supplementation with the intestinal bacterial metabolite SCFA- acetate, which induces upregulation of the autoimmune regulator (AIRE), known to contribute to Treg cell generation. In humans, low maternal serum acetate is associated with preeclampsia and correlates with serum acetate in the fetus. Preeclampsia is a common pregnancy-associated disorder, thought to be associated with maternal gut microbiota, affecting maternal cardiovascular and immune systems, and reduced regulatory T (Treg) cells. Maternal immune changes in preeclampsia are generally mirrored in the fetal immune system. It has been suggested that SCFAs produced by maternal microbiota may regulate both maternal and fetal immune systems during pregnancy [53].

A Danish study has shown that antibiotic use during pregnancy impacted maternal microbiota and their metabolites and was associated with increased occurrence of immune disorders such as immune atopic dermatitis in infants of mothers with atopy and increased risk of asthma in children 2–10 years old [67]. Altered maternal microbiota during pregnancy has been implicated in type 1 diabetes (T1D), inflammatory bowel disease (IBD), and very early onset (VEO)-IBD [68].

### 4.4. Impact of Maternal Microbiota during Pregnancy on the Fetal Endocrine System, HPA Axis

Maternal gut microbiota activates the fetal neuroendocrine HPA pathway, cortisol secretion, and the physiological response to stress by releasing mediators such as proinflammatory cytokines, microbial antigens, and prostaglandins that cross the BBB. Maternal gut microbiota dysbiosis may induce constant hyperactivity of the HPA axis (Figure 1C, [47]).

The fetal HPA axis is essential for the differentiation and maturation of fetal lungs, liver, and CNS. In animals that give birth to mature young (primates, sheep, and guinea pigs), a large proportion of neuroendocrine maturation occurs in utero [17,69,70]. In contrast, neuroendocrine development occurs in the postnatal period in species that give birth to immature young (rats, rabbits, and mice). As a result, fetal/neonatal manipulations impact different stages of neuroendocrine development depending on the species studied [71].

In humans, glucocorticoid receptor (GR) mRNA is present in the adrenal gland by 8–10 weeks of life [72]; little is known about developmental changes in GR expression in the human fetus later in gestation. The human HPA axis is active as early as 11 weeks gestation [73], and the hormonal activity can be detected between 8 and 12 weeks of gestation; corticotropin-releasing hormone (CRH) immunoactivity and bioactivity are detectable in fetal hypothalamic tissue extracts as early as 12–13 weeks gestation, with bioactivity increasing as a function of gestational age [74]. During pregnancy, the fetal hypothalamus and the placenta also produce CRH, which regulates the maturation of the HPA axis and the secretion of adrenocorticotrophin (ACTH). ACTH acting through the growth factors synchronizes fetal adrenocortical growth, angiogenesis differentiation, and steroidogenesis.

Cortisol is vital in maintaining intrauterine homeostasis and maturation of fetal tissue and is de novo synthesized in humans after 28 weeks of gestation. The fetal hypothalamus responds to the acute stress of arterial hypotension by releasing CRH that stimulates the secretion of fetal ACTH, which in turn enhances cortisol secretion [75]. Placental estrogens also influence the fetal HPA axis by facilitating the conversion of active cortisol to inactive cortisone, thus reducing cortisol concentration in the fetus. Recent epidemiological evidence suggests that stress experienced during the fetal period and exposure to corticosteroids can produce long-lasting changes in neural pathways, predisposing to diseases later in life [48].

### 4.5. Impact of Maternal Microbiota during Pregnancy on Fetal Gut and ENS Development

The ENS develops during the fetal period when the cells are derived from the vagal and sacral neural crest. The development of the fetal ENS is regulated by maternal gut microbial metabolites such as SCFAs. SCFAs pass through the placenta and are transmitted to the fetus, where they provide a source of energy, regulate fetal gut epithelium, and program the fetal metabolic system, neural system development, and immune responses [49,50,51]. It has been shown that the cross-placental transfer of microbial metabolites during pregnancy in germ-free mice (GF) is significantly lower than the transfer observed in specific pathogen-free (SPF) mice [76]. Maternal gut dysbiosis has been shown to impair fetal intestine permeability and integrity [59].

More recently, growing evidence supports the direct impact of bacteria exposure on fetal gut colonization [35]. Placenta and amniotic fluid harbor a distinct microbiota characterized by low richness, low diversity, and the predominance of Proteobacteria. Furthermore, shared features between the microbiota detected in the placenta, amniotic fluid, and infant meconium suggest microbial cross-placental transfer and prenatal microbial seeding of the fetal gut. The meconium microbiota has been shown to share more features with the amniotic fluid microbiota than the maternal fecal and vaginal microbiota, suggesting that the meconium microbiota was seeded from multiple maternal body sites and the amniotic fluid microbiota contributed most to the seeding of the meconium microbiota among the investigated maternal body sites [77]. It has been hypothesized that the process of healthy immune maturation guided by microbial contact may begin during fetal life [35].

The development of the digestive system, both in rodents and humans, follows the development of the immune and the ENS. In mice, the intestinal epithelium acquires villi and crypt structure, which in turn leads to restricted epithelial proliferation and villi cytodifferentiation into functional cell types of the small intestine. During that period, one can observe the development of smooth muscle layers surrounding the gut tube and the ENS. In mice, sensory and motor neurons project into the gut around E14. Mice express brush border, crypt, and Paneth cells at low levels around 14 postnatal days [78], and complete intestinal morphogenesis occurs postnatally in mice [79]. It has been shown in mice that while many anatomical and cellular features develop in utero, the ENS becomes functional during the postnatal period [80].

In humans, the digestive tube, a gut precursor, forms during 3rd week of gestation; the crypt formation develops around the 12th week of gestation, and intestinal functions develop by the 24th week of gestation. Notably, the gastrointestinal tract’s development coincides with ENS [81]. ENS is not fully mature in humans and continues development after birth [82]. Enteroendocrine cells emerge as early as the 13th week of gestation. Brush border, crypt, and Paneth cells are expressed at low levels in the fetal intestine at 20 weeks of gestation, and brush border and crypt maturation occur before birth in humans. Tight junction proteins, including the Claudin family of proteins, control the intestinal epithelial barrier; Claudin is expressed in the human intestinal epithelium as early as the 18th week of gestation. The integrity of the intestinal epithelial barrier determines the transport across the lumen and the exclusion of pathogens. Thus, the completion of intestinal morphogenesis occurs prenatally in humans [79].

Mucosal immunity develops in the human fetal intestine by 11–14 weeks gestation. The developing intestine is populated by dendritic cells capable of responding to microbial stimuli and initiating T response. By the 13th gestational week, memory T cells are abundant in the fetal intestine. Specific and highly limited immunomodulatory microbes might be present in the fetal intestine and contribute to fetal immune priming. Bacterial-like morphology was identified in pockets of human meconium at mid-gestation and confirmed by sequencing. Eighteen taxa were enriched in fetal meconium and accompanied by patterns of T cells. Sparse but viable bacteria were observed in the human fetal intestine at mid-gestation with the capacity to limit inflammatory potential by fetal immune cell populations. Fetal T cells are capable of memory formation in the intestine. The presence of bacteria in the fetal intestine suggests that bacterial antigens may also contribute to T cell activation. These specific bacteria persist in nutrient-limiting conditions, grow on pregnancy hormones, and survive within phagocytes [83].

In humans, immunoglobulins are transferred during the fetal period across the placenta, forming one layer between fetal and maternal circulation, which allows for a selective maternal–fetal macromolecular transfer. The transfer of IgG is low during the first and second trimesters but increases during the third trimester [64].

Several laboratories have reported bacterial DNA in the placenta and amniotic fluid. Studies of mice have shown that fetal intestine bacterial DNA is very similar to the placental bacterial DNA but also overlaps with maternal oral and vaginal DNA and meconium [84]. Interestingly, the human fetus begins to produce meconium as early as 12 weeks of gestation, suggesting some fetal–maternal microbiota exchange [85].

## 5. Factors Affecting Maternal Gut Microbiota during Pregnancy: Gender-Specific Multigenerational Effect

Maternal gut microbiota during pregnancy is crucial in programming fetal development, including the components of the GBA. Under healthy conditions, a distinct pregnancy-specific composition of maternal gut microbiota assures optimal development of the fetal brain, VN, immune system, HPA, and gut, which form a rudimentary GBA. However, several factors can alter maternal microbiota during pregnancy besides genetics, such as maternal obesity, diet, maternal stress and depression, infection, the use of medication (antidepressant and antibiotics), and impact the development of fetal GBA [86,87].

### 5.1. Obesity during Pregnancy Is Associated with Changes in Maternal Gut Microbiota (Table 1)

Alterations in maternal gut microbiota during pregnancy accompanied by obesity present a significant health risk to the mother and the fetus and impact the long-term health of the offspring [18,22,88,89].

Differences in the gut microbiota were observed between overweight and normal-weight mice [9]. Differences were also reported in the gut microbiota of pregnant overweight and normal weight expecting mothers and between mothers with excessive weight gain compared with average weight gain during pregnancy. Specifically, maternal overweight was characterized by decreased beneficial bacterial species, such as *Bifidobacterium*, *Blautia*, and *Ruminococcus*, involved in offspring cognitive development [90]. Others reported in obese pregnant women an increase in *Firmicutes*, resulting in a high gut *Firmicutes*-to-*Bacteroidetes* ratio, maternal carbohydrate metabolism, and gut inflammation. Changes in maternal metabolism and gut dysbiosis were associated with an increased risk of fetal macrosomia, preterm birth, and increased offspring BMI [22]. Furthermore, obesity before and during pregnancy has been shown to induce an excessive increase in an inflammatory state, most likely due to microbial translocation of harmful Gram-negative bacteria or bacterial infiltration of the placenta or uterus [88]. Obesity in pregnancy induces metabolic syndrome and altered endocrine functions, which affect placental growth, structure, function, and gene expression. In turn, placental dysfunctions adversely affect fetal development. The placenta is enriched with a complex vascularization for fetal blood supply that requires extensive angiogenesis, and altered metabolome in obese mothers affects placental transcriptosome and placental vascularization. Due to maternal obesity, the placenta stores more lipids and reduces lipid transport required for fetal brain development [89]. Interestingly, a study of pregnant women in southern China showed stability of gut microbiota during pregnancy that was linked to stable dietary patterns and a uniform Chinese population. However, weight increase before the pregnancy and delivery was associated with microbiota changes [86].

**Table 1 microorganisms-11-02199-t001:** Factors involved in maternal gut microbiota dysbiosis during pregnancy and the development of the fetal gut–brain axis: selected clinical studies.

Maternal Conditions during Pregnancy	Maternal Gut Microbiota Dysbiosis	Maternal Health Risks	Fetal/Postnatal Health Issues	References
Obesity	Increased ratio of *Firmicutes*: *Bacteroidetes*.	Excessive inflammation. Translocation of harmful Gram-negative bacteria to the placenta, vascular dysfunction of placenta.	Fetal growth restriction, impacted neurodevelopment.	[88]
	Decrease SCFA producing bacteria.	Metabolic syndrome, low grade inflammation, altered endocrine and placental functions	Impaired placental growth, structure, and function; altered growth and development.	[89]
	High concentration of *Bacteroides*, *Clostridium*, and *Staphylococcus.*	Increase in inflammatory process, fat storage.	Heavier newborns with increased risk of overweight.	[18]
	Increase *Firmicutes*, *Firmicutes* to *Bacteroides* ratio *Actinobacteria*	Chronic proinflammatory state.	Increased fetal macrosomia and preterm birth, increased neonatal body weight and body fat.	[22]
Diet: High fat diet (HFD)	Decreased alpha diversity	Elevated levels of proinflammatory cytokines, placental dysfunctions.	Impaired fetal DA system, increased neural-tube defect; higher risk of depression, anxiety postnatally.	[91]
	Increased, Firmicutes to Bacteroides ratio, decreased alpha diversity	Systemic inflammation, autoimmune diseases	Negative Impact on neurodevelopment, neuronal migration, microglia maturation, loss of BBB, increased inflammation.	[40]
Stress	Altered bacterial diversity and composition.	HPA dysregulation	Dysregulation of fetal HPA axis. Altered offspring’s microbiome composition;	[92,93]
Inflammation	Increase in Proteobacteria	Maternal immune activation (MIA)	Fetal inflammation; neuronal disorders in childhood.	[54]
Infection	Dysbiosis	Increased health risks to the mother	Fetal gut inflammation in utero.	[94]
Antibiotics	Increased alpha diversity; altered microbiota at phylum and genus level.	Altered metabolic activity, increased levels of glucose and insulin	Fetal growth and development.	[95,96,97]
Antidepressants	Altered microbial diversity, composition	Increase in spontaneous abortions and still birth	Increased risk of congenital heart disease.	[97,98]
Age at Conception	Maternal dysbiosis increases with age.	Gestational diabetes, obesity, preeclampsia, Digestive and autoimmune disorders.	Fetal macrosomia, altered development	[12]

### 5.2. Diet Is Vital in Regulating Maternal Gut Microbiota before and during Pregnancy (Table 1)

Maternal gut microbiota during pregnancy profoundly affects fetal development, plays a vital role in fetal programming, and contributes to fetal neurodevelopmental disorders with potentially long-term and multigenerational outcomes [40].

Gut microbiota changes are related to overnutrition and undernutrition [18] and increase offspring’s risk of metabolic and neurological functions in offspring [99]. Especially detrimental is a high-fat diet (HFD) associated with decreased alpha diversity and increased *Firmicutes*-to-*Bacteroidetes* ratio [40]. HFD is linked to maternal obesity, low-grade inflammation, and leaky gut syndrome, resulting in increased translocation of bacteria and bacterial products and inducing systemic inflammation, bacteria overgrowth, and induced immune system activation [40].

HFD before and during pregnancy impairs maternal HPA axis plasticity, increases neural tube defects [100], impacts neurodevelopment during the fetal period [101], and results in depressive-like behavior in adolescent and adult offspring [91]. The Western diet, rich in fat and sugar, has been associated with obesity and low levels of maternal SCFAs related to gut dysbiosis [50]. Consumption of a vegetarian diet during early pregnancy is associated with a distinctive microbial composition involved in the synthesis of fatty acids, lipids, folate [102], and SCFAs, while mothers on a European diet rich in animal protein and lipids show a distinct enterotype I microbiota rich in Bacteroides [16]. Maternal diet is one of the critical factors supplying macro- and micronutrients. During development, deficiency in several micronutrients such as vitamin B, folate, iron, and zinc affect neurodevelopment in animals and humans; deficiency in serum folate [103] and vitamin B12 with an elevated homocysteine level during pregnancy may increase the risk for neural tube defects [104]. Furthermore, Swedish epidemiological studies have suggested that the nutrition of grandmothers was linked to mortality risks in granddaughters [105].

### 5.3. Stress during Pregnancy Disrupts Maternal Gut Microbiota and Impacts Maternal and Fetal Organisms (Table 1)

Stress induces changes in maternal gut bacterial diversity and composition and impacts maternally derived metabolites and substrates necessary for normal development in rodents and humans. In mice, hormonal, endocrine, and immune interactions emerge in pregnant females between gestation days 8–11. Maternal stress during the first week of pregnancy induces rapid, long-lasting, sex-specific alterations in maternal gut microbiota, increased biosynthesis of fatty acids, and degradation of branched-chain amino acids. These changes in maternal microbiota, in turn, result in alterations of placental transfer of nutrients in a sex-specific manner where male but not female offspring demonstrate significant neurodevelopmental changes in the hypothalamic and limbic circuits and the regulation of stress response and may determine cognitive, sensory, and behavioral functions in offspring throughout the life span [106].

In humans, maternal stress has been shown to alter maternal gut microbiota and influence offspring microbiome composition. Prenatal exposure to nutritional insufficiency and stress was associated with increased pathogenic and opportunistic bacteria and decreased health-promoting bacteria [92]. It has been suggested that maternal stress during pregnancy results in HPA dysregulation that may alter the mother’s microbiota, and impact fetal development.

Importantly, the effects of maternal stress during pregnancy pass to the next generations and modulate hippocampal development and gut functions in adult male offspring. Thus, women exposed to stress or depression during pregnancy show dysregulation of the HPA axis. Consequently, the increased circulating cortisol affects maternal gut microbiota. Cortisol can cross the placental barrier, increasing the circulating levels in the fetus and resulting in dysregulation of the HPA axis [93].

### 5.4. Maternal Inflammation, Infection, Antibiotics, and Antidepressants Negatively Affect Maternal Gut Microbiota during Pregnancy (Table 1)

Low-grade systemic inflammation [18] leads to maternal gut microbiota dysbiosis. Inflammation during pregnancy contributes to health issues in both the mother and the fetus [107]. Notably, maternal inflammation during pregnancy leads to vascular dysfunctions of the placenta and fetal growth restriction [88] and affects neurodevelopment.

Infectious and noninfectious diseases also lead to gut microbiota dysbiosis [24,94] and may activate intestinal inflammation in utero due to the fetus swallowing bacterial pathogens and can lead to neurodevelopmental disease in childhood and adulthood [108].

Antibiotic use during pregnancy can lead to microbiota dysbiosis, increasing the risk to the mother and the fetus [95,109]. The use of antibiotics during pregnancy is associated with maternal gut microbiota dysbiosis and altered metabolism, resulting in higher levels of glucose metabolism and insulin [95]. It has been reported that prenatal exposure to norfloxacin and sulfamethoxazole adversely affects fetal growth and development [53].

Likewise, antidepressants—selective serotonin reuptake inhibitors, tricyclic antidepressants, and antipsychotics—negatively affect gut microbiota [110,111,112,113,114]. It has been reported that serotonin and antidepressant fluoxetine impact intestinal colonization of the gut bacterium [115] and affect behavior [116]. SSRI/SNRI antidepressants cross the placenta and BBB and increase the levels of monoamines in the brain [97]. These can then affect the functional development of the brain and the behavior of the child. Antidepressant use by the mother for three months before and during pregnancy can severely increase their children’s risk of congenital heart disease (CHD) [98].

### 5.5. Maternal Age at Conception Impacts Maternal Microbial Diversity (Table 1)

Maternal age at conception has been defined as a significant contributor to microbial diversity in the mother’s gut microbiota, with significant differences across alpha and beta diversity metrics for mothers below and above 30 years of age [12]. Aging is associated with a decline in physiological functions, including GI physiology, and impacts maternal gut microbiota [117]. Maternal gut microbiota dysbiosis in older mothers occurs both during pregnancy and nursing and has been shown to affect infants’ health at two months [12]. Dysbiosis in maternal gut microbiota during pregnancy contributes to health issues in the mother, such as gestational diabetes, obesity, preeclampsia, digestive and autoimmune disorders, fetal health (fetal macrosomia), and development [107].

### 5.6. Multigenerational Effect of Altered Maternal Gut Microbiota during Pregnancy

According to the “Baker’s hypothesis” or DOHaD, prenatal environmental factors influence the offspring’s health in adulthood [118] and have multigenerational effects. Studies in mice demonstrated that maternal high-fat diet (HFD)-dependent insulin resistance impairs synaptic plasticity in the hippocampus and learning and memory in offspring until the third generation [119]. Rats exposed to excess glucocorticoids during fetal development showed reduced birth weight and glucose intolerance into a third generation. Exposure to psychosocial stress during pregnancy affects the grandchildren’s DNA methylation, specifically the DNA methylation of genes involved in stomach organogenesis [105]. Prenatal exposure of mothers to stress has been associated with cardiovascular disease risk. During Dutch Hunger Winter (1944–1945), nutritional deprivation during grandmaternal pregnancy resulted in severe or morbid overweight in grandchildren [105].

## 6. Conclusions

Maternal gut microbiota is uniquely adapted to the pregnancy demands of the mother and the developing fetus. Both animal and human studies underscore the critical association between the composition of maternal microbiota during pregnancy and fetal development, including the formation of rudimentary fetal GBA. The placenta further facilitates the development of fetal GBA, providing maternal–fetal nutritional integration but independently synthesizing hormones and neurotransmitters necessary for fetal development. It is unclear whether the maternal–fetal interaction is limited to the maternal gut metabolites. The emerging evidence suggests that a limited number of unique bacteria infiltrate the fetal milieu and can be identified in amniotic fluid and fetal gut. Independently of whether maternal microbiota regulates fetal development by microbial metabolites translocating to the fetus or/and bacteria infiltrating the placenta, several factors affecting maternal gut microbiota during pregnancy have been identified, including maternal obesity, diet, stress and depression, infection, and drugs (antibiotics and antidepressants) are associated with sex-dependent differences in fetal development. Changes in maternal gut microbiota during gestation have been shown to have a long-term impact postnatally into adulthood and exert multigenerational effects. Thus, understanding the role of maternal gut microbiota during pregnancy in the development of fetal GBA is crucial for managing fetal, neonatal, and adult health.

## 7. Future Perspectives

Given the emerging evidence for some bacteria in amniotic fluid and fetal gut, identifying these early bacteria species in the fetal environment is the top priority. Developing an improved methodology will permit characterization of bacterial species and their developmental impact. Further research into this area may lead to the development of therapeutic intervention strategies to support maternal gut microbiota during pregnancy and decrease the risk of fetal developmental issues. A better understanding of the maternal gut microbiota–placenta–fetal interaction may revolutionize understanding of the diseases’ origins and allow for early diagnosis and therapeutic strategies. From the public health perspective, recognition of maternal health conditions contributing to gut dysbiosis during pregnancy calls for developing more stringent health guidelines regarding diet and weight gain for women planning the pregnancy and pregnant mothers during pregnancy. The impact of maternal stress, depression, and the use of antidepressants during pregnancy may require a reevaluation of preventive maternal medical care. Universal access to medical care during pregnancy should be among health priorities from a health benefits perspective, which is also economically prudent.

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
