# Peer review of "The Impact of Maternal Gut Microbiota during Pregnancy on Fetal Gut–Brain Axis Development and Life-Long Health Outcomes"

_microorganisms, 2023, doi:10.3390/microorganisms11092199_

Round 1

Reviewer 1 Report

In the manuscript "The impact of maternal gut microbiota during pregnancy on fetal gut-brain axis development and life-long health outcomes" the author describes the impact of maternal gut microbiota during pregnancy on the fetal gut-brain axis development. The manuscript is written in great detail, which makes it difficult to read in some parts where the physiological processes that occur during the development of the fetus are described. In certain parts such as the introduction (lines 73-82) and the conclusion and Future perspectives, the text is repeated, so it should be rearranged. Figure 1 is cluttered and should be reorganized (not everything needs to be laid out linearly) and should be linked to the text. The title of the image itself does not comply with the instructions for authors. No description is given below the picture.

The table should also be rearranged according to the instructions for the authors because it is unreadable.

Titles "5.5. Maternal age at conception has been defined as a significant contributor to microbial diversity in the mothers' gut microbiota, with significant differences across all alpha and beta diversity metrics for mothers below and above 30 (Table 1). " and "5.6. Multigenerational effect of altered maternal gut microbiota during pregnancy is consistent with the Developmental Origins of Health and Disease theory of Barker." are inappropriately long and should be shortened.

Author Response

Please see the attached response.

Reviewer 2 Report

This review presents the evidence supporting the prenatal impact of maternal gut microbiota on fetal GBA development.

- The main point of the manuscript is that it is difficult to state that the composition of the maternal microbiota is associated with the development of neurological or psychiatric diseases in children, without first investigating other factors that interfere with the formation of the microbiota, such as diet. Probably the family diet interferes on the maternal microbiota, as well as in the microbiota of the children of the same family. Thus, we cannot state that the maternal microbiota is solely responsible for the development of the babies' CNS. The author needs to discuss the impact of these factors on the baby’s development, together with the microbiota impact.

- The authors provided a well-written review about maternal gut microbiota and the babies’ development, but I missed more information regarding the main gut microbial phyla and genera. I would like to suggest to the authors to add some information about gut microbiota phyla and genera in the main points of the article.  

- The resolution of figure 1 is not good enough to read.

- The resolution of Table 1 is not good enough to read.

Author Response

Please see the attached response.

Round 2

Reviewer 1 Report

I have no further comments.